# Occurrence and Leaching Behavior of Chromium in Synthetic Stainless Steel Slag Containing FetO

Qiang Zeng [1], Jianli Li [1,2,3,*], Yue Yu [1] and Hangyu Zhu [1,*]

1 The State Key Laboratory of Refractories and Metallurgy, Wuhan University of Science and Technology, Wuhan 430081, China; zengqiang@wust.edu.cn (Q.Z.); yuyue@wust.edu.cn (Y.Y.)
2 Hubei Provincial Key Laboratory for New Processes of Ironmaking and Steelmaking, Wuhan University of Science and Technology, Wuhan 430081, China
3 Key Laboratory for Ferrous Metallurgy and Resources Utilization of Ministry of Education, Wuhan University of Science and Technology, Wuhan 430081, China
* Correspondence: jli@wust.edu.cn (J.L.); zhuhy@wust.edu.cn (H.Z.)

**Abstract:** Stainless steel slag has been applied to other silicate materials due to its $CaO$-$SiO_2$-based system. This is done to improve the utilization rate of stainless steel slag and apply it more safely. This paper investigated the occurrence of chromium in synthetic stainless steel slag containing FetO and its leaching behavior. The phase composition of the equilibrium reaction was calculated by FactSage 7.3 Equlib module. XRD, SEM-EDS and IPP 6.0 were used to investigate the phase compositions, microstructure and count the size of spinel crystals. The results indicate that the increase of $Fe_2O_3$ content can promote the precipitation of spinel phases and effectively inhibit the formation and precipitation of $\alpha$-C2S in a $CaO$-$SiO_2$-$MgO$-$Cr_2O_3$-$Al_2O_3$-$FeO$ system. $Fe_2O_3$ contents increased from 2 $wt\%$ to 12 $wt\%$, and the crystal size increased from 4.01 μm to 6.06 μm, with a growing rate of 51.12%. The results of SEM line scanning show the Cr-rich center and Fe-rich edge structure of the spinel phase. Comparing the TRGS 613 standard with the HJ/T 299-2007 standard, the leaching of $Cr^{6+}$ in the FetO samples is far lower than the standards' limit, and the minimum concentration is 0.00791 mg/L in 12 $wt\%$ $Fe_2O_3$ samples.

**Keywords:** stainless steel slag; $Cr^{6+}$; leaching; $Fe_2O_3$; spinel

## 1. Introduction

Stainless steel slag is a $CaO$-$SiO_2$-based slag system, containing parts of $MgO$, $Al_2O_3$, $Cr_2O_3$, $FeO$, $Fe_2O_3$ and other components, and is suitable to be applied to other silicate materials, such as cement, ceramics, glass and refractories [1–5]. Stainless steel slag, as solid waste, has a low utilization rate of its tailings after the metals and slag are separated due to a leaching risk of $Cr^{6+}$ under the temporarily stored treatment [6,7]. Therefore, its comprehensive utilization is of great significance to clean production, resource saving and environmental protection. At present, researchers in the whole world try to use stainless steel slag as part of raw materials to produce other inorganic mineral productions and evaluate the performance and indicators of the products [8,9]. Gensel, O. et al. [10] added 0 $wt\%$, 10 $wt\%$, 20 $wt\%$ and 30 $wt\%$ ferrochrome slag, respectively, to make bricks. The slag-based composition was 0.93 $wt\%$ $CaO$, 29.38 $wt\%$ $SiO_2$, 38.5 $wt\%$ $MgO$, 23.47 $wt\%$ $Al_2O_3$, 5.17 $wt\%$ $Cr_2O_3$ and 1.55 $wt\%$ $Fe_2O_3$. The dried samples were heated to 900 °C at rate of 5 °C/min under the pressure of 20 MPa after the treatment by semi-dry mixtures. All of the mineral phases, $(Mg, Fe)_2SiO_4$, $MgAl_2O_4$ and $MgSiO_3$, are high temperature precipitated phases. EDS results showed that the slag system of 30 $wt\%$ ferrochrome brick was 1.12 $wt\%$ $CaO$, 50.53 $wt\%$ $SiO_2$, 8.07 $wt\%$ $MgO$, 23.99 $wt\%$ $Al_2O_3$, 1.34 $wt\%$ $Cr_2O_3$ and 10.23 $wt\%$ $Fe_2O_3$. $SiO_2$, spinel and $(Mg, Fe)_2SiO_4$ were the main mineral phases in the sample. The porosity and thermal conductivity of all bricks met the standard of building insulation materials. In the production process of bricks, Cr was considered to exist in $Cr_2O_3$, and its content in the final product was 1.34 $wt\%$.

Yang et al. [11] prepared a glass ceramics with the addition of stainless steel slag into a kind of pre-melted slag, and the results revealed that the $Fe_2O_3$ content ranges from 5.4 to 5.6 *wt*%, and the $Cr_2O_3$ contents were 0 *wt*%, 0.2 *wt*%, 0.5 *wt*%, 0.7 *wt*% and 0.9 *wt*%, respectively. According to the solid waste leaching standard HJ/T299-2007 (China), the leaching amount of $Cr^{6+}$ was 0.007 mg/L, which was lower than the standard of 5 mg/L specified in GB 5085.3-2007 (China). $Cr_2O_3$ was stabilized in the crystalline phase as a nucleating agent, and the formation of the glass phase further prevented the leaching of $Cr^{6+}$ after the glass ceramics were doped with stainless steel slag [11]. Yeong found that 100 *wt*% replacement hardened concrete was better and the leaching amount of $Cr^{6+}$ was lower than the standard requirement after 0 *wt*%, 25 *wt*%, 50 *wt*%, 75 *wt*% and 100 *wt*% stainless steel slag were replaced the original aggregate to determine the optimal replacement ratio. Thus, stainless steel slag can be used as a non-toxic concrete additive [12].

For a high temperature remelting process, it is necessary to consider the influence of products basicity, compositions and subsequent cooling regimes on the phase compositions of the system in order to reduce the leaching toxicity of the products [13]. The temperature and atmosphere under processing conditions of the products should be considered during the service period [14] in order to improve the utilization rate of stainless steel slag and explore the stabilization method. TCLP (toxicity characteristic leading procedure) as a heavy metal pollution assessment method is implemented by the USA; in Germany it is TRGS 613 (technical rules of hazardous substances) and the Chinese solid waste leaching standard is HJ/T299-2007 [15–17]. $CaO/SiO_2$ (basicity) has a great influence on the phase composition. Zhao et al. [18] studied the distribution of Cr in 1.0, 1.5 and 2.0 basicity samples. It was found that all of Cr was in the glass matrix at 1.0 basicity and 1600 °C. About 54.3% of Cr exists in spinel phase, 3.1% of Cr exists in C2S and the leaching amount of $Cr^{6+}$ decreases from 2.28 mg/L to 2.26 mg/L as the basicity is 1.5. The enrichment rate of Cr in spinel phase reaches 61.7% as the basicity is 2.0, and the rest of Cr is dissolved in the periclase phase. This phase is unstable in an acid solution and the leaching amount is 3.68 mg/L. The highest enrichment of Cr in spinel phases is 91.2%, and the lowest leaching amount is 0.62 mg/L at 1300 °C. Wu et al. [19] investigated the effects of basicity and FeO on the crystallization behavior of Cr in stainless steel slag. It was found that the slag containing FeO was easier to form an amorphous phase at low basicity (B = 1.0), and spinel crystal and merwinite were formed in the samples with higher basicity (B = 1.25 or 1.5). The results also showed that lower basicity (1.2) and higher FeO content were favorable for the formation of $Mg(Al, Fe, Cr)_2O_4$ from $Cr_2O_3$. The cooling regimes were to prevent the transformation of α-C2S with a density of $3.28 \times 10^3$ $kg/m^3$ to γ-C2S with the density of $2.97 \times 10^3$ $kg/m^3$ to suppress volume expansion of stainless steel slag [20]. In general, the purposes of changing basicity and the compositions are to adjust the phase compositions and then suppress the leaching of $Cr^{6+}$.

The effect of FeO on spinel crystallization and chromium stability in $CaO$-$SiO_2$-$MgO$-$Al_2O_3$-$Cr_2O_3$ was studied in the previous work [21]. The results indicated that spinel crystal $(Mg, Fe) (Al, Fe, Cr)_2O_4$ formed an Fe-rich shell structure. The leaching amounts were 0.1434 mg/L and 0.0021 mg/L, and the FeO contents were 0 *wt*% and 20 *wt*%, respectively. Although the growth of spinel is beneficial to enrich the Cr in spinel crystals, there are defects in the huge spinel crystals, which may induce the transformation of $Cr^{3+}$ to $Cr^{6+}$. Mou et al. [22] studied the effects of $Fe_2O_3$ on the $CaO$-$SiO_2$-$MgO$-$Al_2O_3$-$Cr_2O_3$ system and found that the addition of $Fe_2O_3$ raised the content of the liquid phase. At the same time, the growth of spinel crystals was promoted. In Sørensen's research [23], $Fe^{2+}$ had stronger mobility, while $Fe^{3+}$ as a network-forming agent reduced the mobility of $Mg^{2+}$ and $Ca^{2+}$. Fe in the stainless steel slag mainly exists in the form of $(FeO, Fe_2O_3)$. Therefore, it is of great practical significance to study the effect of FetO on Cr leaching. In this paper, $Fe_2O_3$ is added as extra content in the $CaO$-$SiO_2$-$MgO$-$Al_2O_3$-$Cr_2O_3$-8 *wt*% FeO system.

## 2. Experimental

### 2.1. Samples Preparation

The slag samples were prepared according to the components shown in Table 1. Samples were prepared from analytically pure chemical reagents and using $FeC_2O_4$ as a source of FeO generated from thermal decomposition. Among them, $CaO$-$SiO_2$-$MgO$-$Al_2O_3$ was 100 g, and FeO and $Fe_2O_3$ were extra added by weight percentage. The slag sample was accurately weighed, mixed evenly and charged into a corundum crucible, which was put into a carbon-tube furnace (25 kw, 1650 °C) as shown in Figure 1. The furnace was heated to 1550 °C with the rate of 5 °C/min below 300 °C and 10 °C/min above 300 °C under an argon atmosphere. After holding for 30 min at 1550 °C, the crucible was taken out and air cooled. The cooled samples were crashed, and a selected portion was encased in resin (HMR4) with thermal mosaic (XQ-2B). The sample microstructure was observed by SEM-EDS (NanoSEM400, FEI, Hillsborough, OR, USA), and the composition of the micro-area was analyzed. After ball milling of the slag, a powder sample of 2 g was taken and used for XRD analysis (X Pert Pro MPD, Malvern Panalytical Ltd., Malvern, UK).

**Table 1.** Compositions of synthetic slag sample/g.

| No. | CaO | $SiO_2$ | MgO | $Al_2O_3$ | $Cr_2O_3$ | FeO | $Fe_2O_3$ | B |
|-----|-------|-------|------|------|------|------|-------|------|
| C1 | 46.67 | 33.33 | 8.00 | 6.00 | 6.00 | 8.00 | 2.00 | 1.40 |
| C2 | 46.67 | 33.33 | 8.00 | 6.00 | 6.00 | 8.00 | 5.00 | 1.40 |
| C3 | 46.67 | 33.33 | 8.00 | 6.00 | 6.00 | 8.00 | 8.00 | 1.40 |
| C4 | 46.67 | 33.33 | 8.00 | 6.00 | 6.00 | 8.00 | 12.00 | 1.40 |

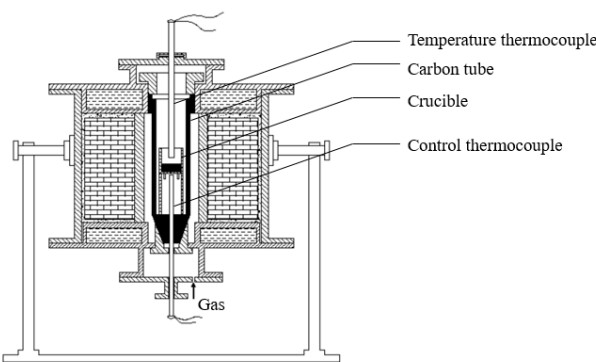

**Figure 1.** Experimental device diagram.

### 2.2. Leaching Test

According to TRGS 613 leaching standard, $Cr^{6+}$ leaching test was carried out. Accordingly, a 1 g sample, 20 mL HCl (1 mol/L) and deionized water were put into a 250 mL beaker, and the volume of the solution was 200 mL, which was stirred with electromagnetic stirrer for 15 min (rotor length was 40 mm, rotating speed was 1000 rpm), then filtered with sand core filter (0.45 μm microporous membrane). The indicator was prepared by using 50 mL acetone (analytical pure) with 0.5 g diphenylcarbazide and one drop of acetic acid (analytical pure). Then, a 1 mL indicator, 1 mL hydrochloric acid (1 mol/L), 2 mL leaching liquid and deionized water were used to prepare the test solution, and the volume was fixed with a 50 mL volumetric flask. The spectrophotometer was used to measure the spectrophotometry at 540 nm. According to the standard curve, the concentration of $Cr^{6+}$ in leaching solution was calculated. After the first standard leaching experiment, the filter residue was continuously leached according to TRGS 613 standard until the leaching solution became colorless; thus, the total amount of $Cr^{6+}$ was able to be obtained.

### 2.3. Thermodynamic Calculation

The phase composition of the equilibrium reaction was calculated according to Fact-Sage 7.3 Equlib module. The calculation conditions were set as follows:

(1) Units: temperature/°C, pressure/atm, mass/g.
(2) Databases: FactPS, Ftoxid.
(3) Compounds: ideal-pure solids.
(4) Solutions: FToxid-(SLAGA, SPINA, MeO_A, bC2SA, aC2SA, Mel_A, CaSpinel).

## 3. Results and Discussion

### 3.1. Calculation and Analysis of Slag Components

The effect of $Fe_2O_3$ contents on the phase composition of $CaO$-$SiO_2$-$MgO$-$Al_2O_3$-$Cr_2O_3$-8 *wt*% FeO system at 1550 °C was calculated by FactSage 7.3. The results showed that $Mg_2SiO_4$, $Fe_2SiO_4$ and α-C2S were the same mineral phases in tbe 2 *wt*% and 5 *wt*% $Fe_2O_3$ samples, and the spinel crystals were $MgCr_2O_4$, $FeCr_2O_4$, $MgAl_2O_4$ and $FeAl_2O_4$. The 8 *wt*% and 12 *wt*% $Fe_2O_3$ contents had the same mineral phases except for α-C2S. Figure 2 shows the variation of the theoretical precipitation amount of spinel crystal and α-C2S with $Fe_2O_3$ contents at 1550 °C. When the $Fe_2O_3$ content increased from 2 *wt*% to 12 *wt*%, the precipitation amount of spinel crystal in $CaO$-$SiO_2$-$MgO$-$Al_2O_3$-$Cr_2O_3$8 *wt*% FeO system varied from 7.87 g to 8.29 g at 1550 °C. The precipitated amount of α-C2S gradually decreased from 13.99 g to 0 g at 1550 °C. The C2S was precipitated as in Equation (1). Table 2 shows the activity of components calculated by FactSage 7.3 at 1550 °C, and the free Gibbs Energy variation ΔG is the negative value, which indicates that the reaction can proceed spontaneously in this state. The reaction equilibrium constant decreases with the increase of $Fe_2O_3$ addition, indicating that the degree of spontaneous reaction becomes weaker.

$$(CaO)+2(SiO_2)= Ca_2SiO_4(s) \ \Delta G = -RT \ln K \tag{1}$$

$$K = \frac{a(Ca_2SiO_4)}{a(CaO) \cdot a^2(SiO_2)} \tag{2}$$

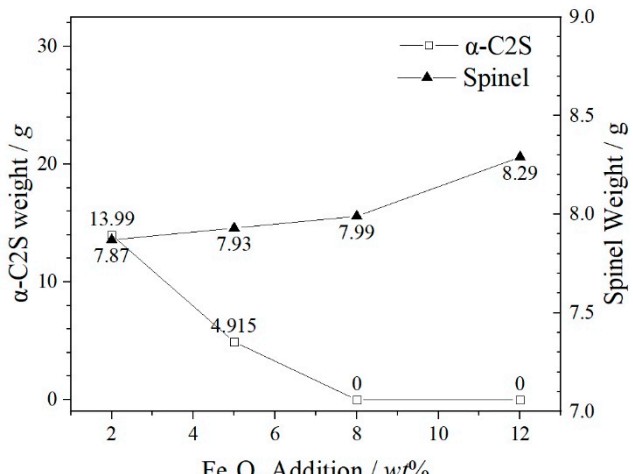

**Figure 2.** Amount of a-C2S and spinel phases (g) in the slag samples at 1550 °C versus $Fe_2O_3$ addition (%).

**Table 2.** Activity calculation of FactSage 7.3 at 1550 °C.

| Activity | 2 *wt*% | 5 *wt*% | 8 *wt*% | 12 *wt*% |
|---|---|---|---|---|
| CaO | 0.0098 | 0.0096 | 0.0091 | 0.0081 |
| SiO$_2$ | 0.0104 | 0.0111 | 0.0120 | 0.0138 |
| α-C2S | 0.8049 | 0.8141 | 0.7947 | 0.7319 |
| K | $7.58 \times 10^5$ | $6.88 \times 10^5$ | $6.05 \times 10^5$ | $4.70 \times 10^5$ |

The most prominent strength peaks of all samples are in a similar position from Figure 3 and fitted by spinel crystals. Similar XRD patterns indicate that the main phase compositions in 5 *wt*%, 8 *wt*% and 12 *wt*% Fe$_2$O$_3$ samples are coincident. There is the diffraction peak of α-C2S in the 2 *wt*% Fe$_2$O$_3$ sample, and the α-C2S peak disappears in the 5 *wt*% Fe$_2$O$_3$. Compared with Figure 2, the calculated result is kind of different from the XRD result in the 5 *wt*% Fe$_2$O$_3$ sample, which produces 4.91 g α-C2S at 1550 °C, as seen in Figure 2. It is explained by the liquid phase ratio increasing with the increase of the Fe$_2$O$_3$ content in the samples and the small amount of α-C2S precipitated in the system. The α-C2S cannot nucleate and grow up under the control of cooling conditions and exists as silicate matrix. As shown in Figure 4, the increase of Fe$_2$O$_3$ content significantly increases the proportion of the liquid phase.

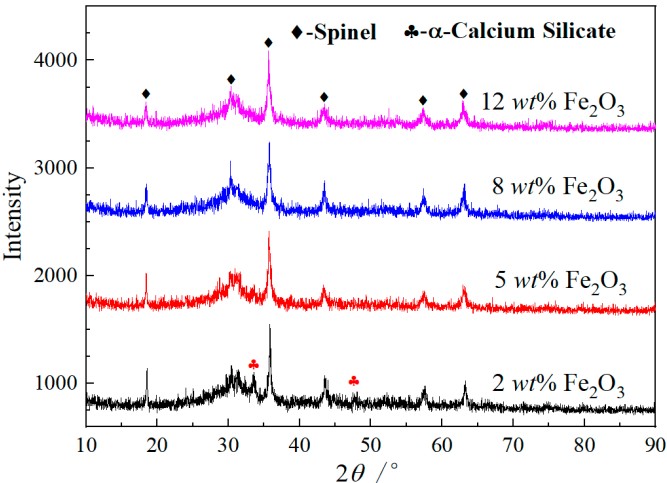

**Figure 3.** XRD patterns of steel slag samples.

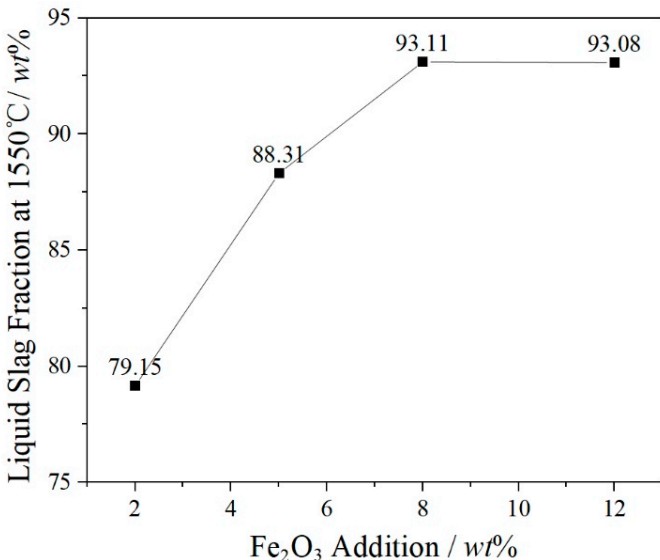

**Figure 4.** Effect of Fe$_2$O$_3$ on liquid content in CaO-SiO$_2$-MgO-Al$_2$O$_3$-Cr$_2$O$_3$ system at 1550 °C.

### 3.2. Microstructure

The effects of Fe$_2$O$_3$ on the microstructure of CaO-SiO$_2$-MgO-Al$_2$O$_3$-Cr$_2$O$_3$-8 *wt*% FeO system are shown in Figure 5. Three distinct phases, namely spinel crystals, dicalcium silicate and silicate matrix, are in the 2 *wt*% Fe$_2$O$_3$ sample (Figure 5a). There are the only spinel and matrix phases as the Fe$_2$O$_3$ content are higher than 2 *wt*%, and the dicalcium silicate disappears. Based on the SEM, 10 points of spinel crystals in each sample in Figure 5 were analyzed by EDS. The average compositions and element distribution are shown in Table 3 and Figure 6.

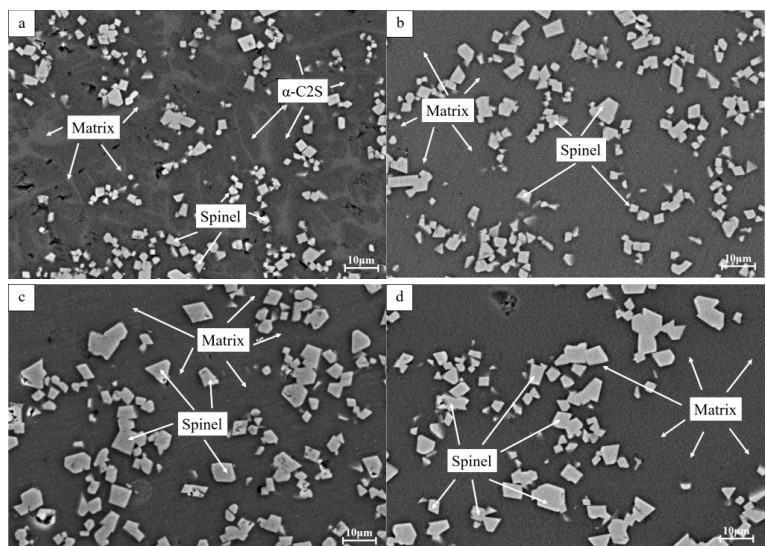

**Figure 5.** Effects of Fe$_2$O$_3$ on the microstructure of the CaO-SiO$_2$-MgO-Al$_2$O$_3$-Cr$_2$O$_3$-8 *wt*% FeO system ((**a**)—2 *wt*%, (**b**)—5 *wt*%, (**c**)—8 *wt*% and (**d**)—12 *wt*%).

**Table 3.** Chemical composition of spinel crystals/atom%.

| Fe$_2$O$_3$ | O | Mg | Al | Si | Ca | Cr | Fe |
|---|---|---|---|---|---|---|---|
| 2 *wt*% | 51.18 | 9.89 | 4.57 | 2.73 | 3.02 | 19.60 | 9.03 |
| 5 *wt*% | 54.37 | 9.46 | 3.56 | 2.57 | 2.80 | 16.35 | 10.91 |
| 8 *wt*% | 54.30 | 10.33 | 4.02 | 2.41 | 2.49 | 14.37 | 12.09 |
| 12 *wt*% | 54.01 | 9.85 | 3.83 | 1.15 | 1.39 | 12.79 | 16.99 |

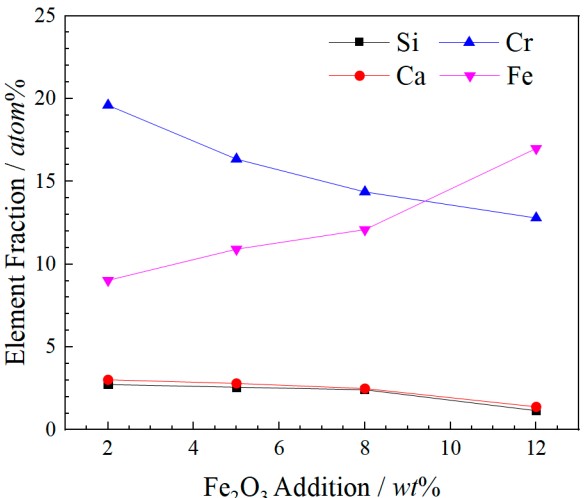

**Figure 6.** Variation of spinel crystals' chemical compositions.

The positions of spinel phase in Figure 5 show the spinel phase is mainly formed in the whole dispersion and local aggregation, which is related to the diffusion of solute particles and the nucleation of core components. According to Figure 4, the solute liquid phase content increases with the increase of $Fe_2O_3$ content. The liquid phase content of 12 *wt*% $Fe_2O_3$ sample is slightly lower than that of 8 *wt*% $Fe_2O_3$ sample, which is 93.08%. More content of liquid solute is beneficial to crystal growth. The spinel phase in Figure 5a is mostly embedded by the α-C2S phase, while the spinel phase in Figure 5b–d) is embedded in the silicate matrix. Fe content and the content of Cr, Ca and Si in the spinel crystals are negatively correlated. As the addition of $Fe_2O_3$ in the $CaO-SiO_2-MgO-Al_2O_3-Cr_2O_3$-8 *wt*% FeO system increases from 2 *wt*% to 12 *wt*%, the contents of Cr, Ca and Si decrease from 19.60 at% to 12.79 at%, 3.02 at% to 1.39 at% and 2.75 at% to 1.15 at%, respectively, while Fe increases from 9.03 at% to 16.99 at%. Compared with the spinel phase composition calculated by FactSage 7.3 in Section 3.1, there is no Ca spinel phase in the experimental samples. A minute quantity of Ca was detected by EDS; thus, $Ca^{2+}$ may be substituted into spinel crystal and Si may be the form of an interstitial solution in spinel crystal. Thus, the chemical formula of spinel crystal suggested is $(Mg, Fe) (Cr, Fe, Al)_2O_4$.

It can be seen from Figure 5 that the maximum diameter of spinel crystal gradually increases with the increase of $Fe_2O_3$ content. The size of spinel crystal was counted by IPP 6.0 software as shown in Figure 7. $Fe_2O_3$ contents and the size of spinel crystals are positively correlated; as $Fe_2O_3$ contents increase from 2 *wt*% to 12 *wt*%, the crystal size increases from 4.01 μm to 6.06 μm, with a growing rate 51.12%. Figure 6 shows that the content of Fe and Cr in spinel crystal are negatively correlated, and the reduction amount of Fe and Cr is similar. The sum of their atomic ratios is from 26.48 at% to 29.78 at%. After the atom distribution of 2 *wt*% to 12 *wt*% $Fe_2O_3$ samples in Table 3 on the basis of the combined valence of the spinel solid solution $(Mg, Fe, Ca) (Cr, Fe, Al)_2O_4$, it was found that the combined valence of Fe in FetO is +2.75, +2.71 and +2.26 (except 2 *wt*%), respectively. The lower the valence, the greater the proportion of Cr in the cationic $[(Cr, Fe, Al)_2O_4]^{2+}$ of the spinel solid solution. Therefore, the higher the enrichment degree of Cr in the spinel solid solution, the less the leaching risk of $Cr^{6+}$.

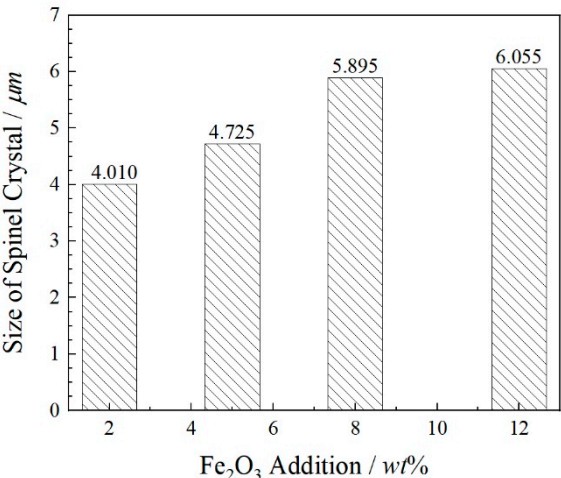

**Figure 7.** Variation behavior of spinel crystal size in samples.

As shown in Figure 8, the spinel crystal was scanned by SEM with 12 *wt*% $Fe_2O_3$ in $CaO-SiO_2-MgO-Al_2O_3-Cr_2O_3$-8 *wt*% FeO system. The spinel crystal has a layered structure, and the white outer layer is Fe rich, and the darker part in the spinel center is Cr rich. The line scanning results of spinel crystal also verify this result. By observing the distribution of Fe and Cr in the spinel crystal region, it can be found that the content of Fe at the phase interface between spinel crystal and silicate matrix is the largest, while the content of Cr at the center of spinel interval is the largest, and Cr is embedded by the Fe-rich outer layer. Rezani [24] studied the effect of $Cr_2O_3$ on the crystallization properties of $CaO-SiO_2-MgO-$

$Al_2O_3$ glass ceramics. The research showed that $Cr_2O_3$ as a nucleating agent was beneficial to the crystallization process of melting, and $Fe_2O_3$ could strengthen the crystallization process. In the study, $Fe_2O_3$ seems to act as viscosity reducer, and $Fe_2O_3$ as network modifier. In terms of the crystallization process of the system, a growth model of the spinel solid solution with $MgCr_2O_4$ as the initial crystal nucleus was proposed. It is consistent with the growth model of spinel crystals put forward by the author [21]. The line scanning results show that the closer to the nucleation core of spinel crystals, the content of Cr is highest, and the content of Fe at center is lower than that of at boundary. As MgO, FeO and $Cr_2O_3$ exist at the same time, the initial crystal nucleus is more likely to be $(Mg, Fe)_2CrO_4$, and then the crystals grow on this basis. $Fe_2O_3$ activity at the spinel boundary of rises with the increase of $Fe_2O_3$ content, and the decrease of viscosity is conducive to particle diffusion, which further promotes the growth of spinel. Therefore, the structure of the Cr-rich center and Fe-rich edge is formed.

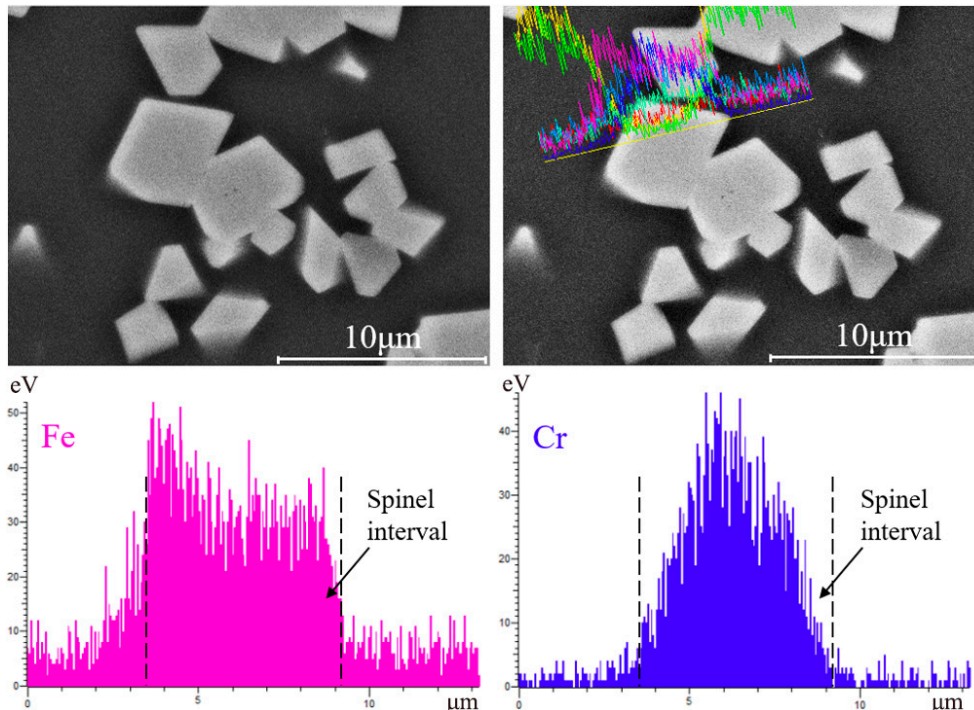

**Figure 8.** Distribution of Fe and Cr elements in spinel crystal in the 12 *wt*% sample.

### 3.3. Leaching Toxicity

Figure 9 shows the color of the leaching solutions and their concentration of $Cr^{6+}$ after the leaching test of the stainless steel slag samples C1–C4 according to TRGS 613 standard. The concentration of $Cr^{6+}$ in the leaching solution was calculated from the fitted equation obtained from the calibration solution. The lowest concentration of calibration solution is 0.02 mg/L. The Equation is described below.

$$c = n \times (N - 0.0014)/0.8529 \tag{3}$$

where $c$ is the concentration of $Cr^{6+}$ in mg/L, n is the dilution ratio and $N$ is the measured value of spectrophotometer.

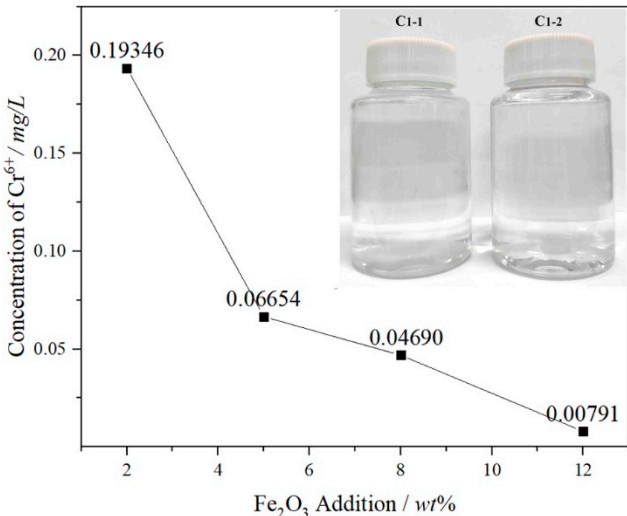

**Figure 9.** Color of leaching solutions and concentration of $Cr^{6+}$.

According to TRGS 613 standard, as shown in Figure 9, the maximum concentration of $Cr^{6+}$ in the leaching solution is 0.19346 mg/L. Yellow is the characteristic color of $Cr^{6+}$ leaching solutions, and four samples are clarified and transparent, which indicates that the amount of $Cr^{6+}$ in the leaching solution is minimal. Nitric acid and sulfuric acid are used as the extraction solution in the HJ/T 299-2007 standard, and the pH of the solution sis $3.20 \pm 0.05$. Hydrochloric acid is the extraction solution (pH = 1) in the TRGS 613 standard. Acid solution can accelerate the dissolution of hydroxide and silicate phase, thus releasing Cr. Studies show that $Fe^{2+}$ can reduce $Cr^{6+}$ to $Cr^{3+}$ under acidic conditions (Equation (4)) [25]. In these leaching experiments, the leaching concentrations of $Cr^{6+}$ decreased with the increase of the $Fe_2O_3$ content. The reason is that the $Fe_2O_3$ content improves the size of the spinel crystals and enriches the Cr in the spinel crystals.

The Fe-rich shell structure of the spinel crystals can control the leaching of $Cr^{6+}$ well. It also shows that the leaching concentration of $Cr^{6+}$ in the FetO slag system is far lower than the standard limit (5 mg/L), and the minimum concentration is 0.00791 mg/L in the 12 *wt*% $Fe_2O_3$ sample.

$$Cr_2O_7^{2-}+6Fe^{2+}+14H^+ \rightarrow 2Cr^{3+}+6Fe^{3+}+7H_2O \tag{4}$$

## 4. Conclusions

(1) The increase of $Fe_2O_3$ content can promote the precipitation of spinel phases and effectively inhibit the formation and precipitation of $\alpha$-C2S. $Fe_2O_3$ promotes the spinel crystal precipitations as a result of the increase of $FeCr_2O_4$, $MgFe_2O_4$, $MgAl_2O_4$ and $FeAl_2O_4$ in the spinel solid solution by FactSage 7.3. $MgFe_2O_4$ and $Fe_2O_3$ play a key role in the formation of the spinel solid solution $(Mg, Fe) (Cr, Fe, Al)_2O_4$.

(2) $Fe_2O_3$ contents increase from 0 *wt*% to 12 *wt*%, the sizes of spinel crystals increase from 4.01 μm to 6.06 μm and the growth rate is 51.12% compared with these two contents of $Fe_2O_3$. The atomic ratios Fe and Cr are from 26.48 at% to 29.78 at%. The combined valence of Fe in FetO is +2.75, +2.71 and +2.26, respectively, in the 2 *wt*%–12 *wt*% $Fe_2O_3$ samples. The enhancement in the proportion of Fe in the cationic of spinel solid solution promotes the enrichment of Cr in spinel phases and reduces the leaching risk of $Cr^{6+}$.

(3) $Cr_2O_3$ is a nucleating agent to promote the nucleation of the spinel solid solution phase. The increase of the concentration of FetO in the liquid phase promotes the concentration gradient of $Fe_2O_3$ and FeO components on the liquid side of the interface, and the amount of liquid phase is increased. The diffusion conditions of particles are improved, and the structure of spinel phase is a Cr-rich center and an Fe-rich edge.

(4)　The leaching amount of $Cr^{6+}$ in the FetO samples is far lower than the standard limit under the acid leaching conditions by the TRGS 613 standard or HJ/T 299-2007 standard, and the color of the leaching solution is transparent, with the maximum leaching concentration of $Cr^{6+}$ at 0.19346 mg/L in the 2 *wt*% $Fe_2O_3$ sample.

**Author Contributions:** Conceptualization, J.L. and Q.Z.; methodology, Y.Y.; software, Q.Z.; writing—review and editing, Q.Z.; project administration, H.Z.; funding acquisition, J.L. All authors have read and agreed to the published version of the manuscript.

**Funding:** This research was funded by the National Natural Science Foundation of China, grant number 51974210, Hubei Provincial Natural Science Foundation, grant number 2019CFB697.

**Acknowledgments:** The authors would like to thank the State Key Laboratory of Refractories and Metallurgy for the support.

**Conflicts of Interest:** The authors declare no conflict of interest.

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
