# Peer review of "Occurrence and Leaching Behavior of Chromium in Synthetic Stainless Steel Slag Containing FetO"

_minerals, doi:10.3390/min11101055_

Round 1
Reviewer 1 Report
Structure of English should be improved to make the reading more fluid.
The title and abstract should include "synthetic slag" as no slag is being investigated in this study.
Please give quantitative references for utilization of stainless steel slags, because in Europe it is used so the statement that most of it is landfilled is not true, at least in Europe. Steelworks are continuously working to make sure Cr is not leaching from their slag by forming spinel phases so that the slag can be used. Please also give examples of what the stainless steel slag is used for currently, as it is not used in the cement industry. But finding a way to use stainless steel in the steel industry would be interesting for the finer fractions.
In the introduction the first study describes ferrochrome slag which is very different from stainless steel slag.
Use of Al2O3 crucibles for stainless steel remelting is not ideal as the alumina dissolves into the slag during the test, changing the chemistry of the system.
Author Response
Dear Editor and Reviewers:
Thank you very much for your useful comments and suggestions on our manuscript. We have modified the manuscript accordingly, and several figures have been remade. The detailed corrections are listed below point by point.

Reviewer 2 Report
This manuscript deals with the experimental study of synthetic stainless-steel slag after being dosed with varying amounts of Fe2O3 and melted in a furnace under an argon atmosphere. Its aim was to study in the final mixed slag, the distribution of Cr in different phases and Cr leachability under standard leaching testing.
In my opinion this manuscript is of interest as it offers an interesting technical possibility to minimize the risk of heavy metal leaching from some metallurgy slags that are currently discarded and subsequently exposed to ambient conditions.
Remarks:
1.- I understand from the manuscript that the justifying reason of this study is the potential utilization (or treatment) of stainless-steel slags that at present are generally stacked or landfilled with an implicit risk Cr6+ leaching under ambient conditions. Hopefully the potential risk of Cr6+ leaching could be minimized after being treated with a specific Fe2O3 dose. From this research results, a potential application is implied for stainless-steel slags before being disposed. Brief comments related to a possible development of a novel technology to minimize Cr6+ leaching from real slags are encouraged.
2.- What kind of problems would be expected for a similar research but using real slags?
3.- The manuscript has many grammatical errors and confusing sentences.
4.- Many errors and remarks are highlighted in yellow and sticky notes in an attached pdf file of the manuscript edited in adobe acrobat software.
In the present form this work is inadequate for publishing. In my opinion an extensive work must be done before publication.

Author Response

(The authors gave the same response as above.)
